# Intrathecal delivery of adipose-derived mesenchymal stem cells in traumatic spinal cord injury: Phase I trial

Mohamad Bydon [1,2] ✉, Wenchun Qu[3], F. M. Moinuddin[1,2], Christine L. Hunt[4], Kristin L. Garlanger[5], Ronald K. Reeves[5], Anthony J. Windebank[6], Kristin D. Zhao[5], Ryan Jarrah[1,2], Brandon C. Trammell[1,2], Sally El Sammak[1,2], Giorgos D. Michalopoulos [1,2], Konstantinos Katsos[1,2], Stephen P. Graepel[2], Kimberly L. Seidel-Miller[5], Lisa A. Beck[5], Ruple S. Laughlin [6] & Allan B. Dietz [7]

Intrathecal delivery of autologous culture-expanded adipose tissue-derived mesenchymal stem cells (AD-MSC) could be utilized to treat traumatic spinal cord injury (SCI). This Phase I trial (ClinicalTrials.gov: NCT03308565) included 10 patients with American Spinal Injury Association Impairment Scale (AIS) grade A or B at the time of injury. The study's primary outcome was the safety profile, as captured by the nature and frequency of adverse events. Secondary outcomes included changes in sensory and motor scores, imaging, cerebrospinal fluid markers, and somatosensory evoked potentials. The manufacturing and delivery of the regimen were successful for all patients. The most commonly reported adverse events were headache and musculoskeletal pain, observed in 8 patients. No serious AEs were observed. At final follow-up, seven patients demonstrated improvement in AIS grade from the time of injection. In conclusion, the study met the primary endpoint, demonstrating that AD-MSC harvesting and administration were well-tolerated in patients with traumatic SCI.

Spinal cord injury (SCI) is a debilitating condition that carries severe medical, psychological, and financial implications for those living with residual impairments. As of 2016, there are over 290,000 individuals in the United States living with an SCI at a rate of 17,000 new cases per year[1]. Currently available interventions for patients with subacute and chronic SCI are limited to symptomatic management and physical rehabilitation[2]. This reflects the complexity of SCI pathophysiology that occurs in the injured spinal cord. However, there has been an increasing interest in the application of regenerative medicine in spinal cord injury[3]. Stem cell therapy is a prime example of regenerative medicine applications, whereby the interaction between these cells promotes a potential regenerative environment following spinal cord injury[3–5].

Adipose tissue represents the most prominent reservoir of mesenchymal stem cells, named adipose-derived mesenchymal stem cells (AD-MSCs). Adipose-derived MSCs are considered attractive options due to their availability, ease of access, and multipotency[6]. The use of AD-MSC has been thoroughly investigated in traumatic and degenerative diseases[7–11]. Preclinical trials in SCI animal models have found evidence that AD-MSCs can regulate the inflammatory response and promote a regenerative environment, correlating with promising clinical outcomes[12–15].

---

[1]Neuro-Informatics Laboratory, Mayo Clinic, Rochester, MN, USA. [2]Department of Neurological Surgery, Mayo Clinic, Rochester, MN, USA. [3]Physical Medicine and Rehabilitation, Mayo Clinic, Jacksonville, FL, USA. [4]Department of Pain Medicine, Mayo Clinic, Jacksonville, FL, USA. [5]Physical Medicine and Rehabilitation, Mayo Clinic, Rochester, MN, USA. [6]Department of Neurology, Mayo Clinic, Rochester, MN, USA. [7]Department of Laboratory Medicine and Pathology, Mayo Clinic, Rochester, MN, USA. ✉e-mail: bydon.mohamad@mayo.edu

Previously, we presented the case report of a patient with SCI who received intrathecal AD-MSCs; the patient demonstrated significant motor and sensory improvements after a period of neurological plateau[16]. This patient was participating in CELLTOP (Clinical Trial of Autologous Adipose-Derived Mesenchymal Stem Cells in the Treatment of Paralysis Due to Traumatic Spinal Cord Injury; ClinicalTrials.gov Identifier: NCT03308565), a Phase I clinical trial investigating the safety, feasibility, and biological effects of AD-MSC injection in ten patients with traumatic SCI.

Herein, we present the results of the CELLTOP trial for all 10 patients enrolled and followed up for 96 weeks. We demonstrate the safety profile of autologous AD-MSCs harvesting and intrathecal administration in spinal cord injury. During the study period, no serious adverse events were reported, and 7 patients experienced motor and/or sensory improvement.

## Results

### Baseline Characteristics

Ten patients were enrolled, administered AD-MSCs intrathecally, and observed for two years. The total duration of the study was from January 2018 to October 2021. The average patient age was 34.6 years (range 18–65 years). In the cohort, 80% ($n = 8$) of the patients were males, and 20% ($n = 2$) were females. The majority (70%) of the patients were white, followed by black (10%), Hispanic/Latino (10%), and Asian (10%). All patients were enrolled within 12 months of injury (range 2–12 months). Six of the patients had a cervical-level injury, and four had a thoracic-level injury. At the time of injury, eight patients were classified as AIS grade A and two patients as AIS grade B. Motor vehicle accidents (motorcycle, all-terrain vehicle, or automobile) comprised 50% of the SCI etiology ($n = 5$), followed by fall-related injuries ($n = 3$), pedestrian-car accidents ($n = 1$), and surfing accidents ($n = 1$). The average duration between injury and AD-MSC injection was 12 months (range of 7–22). At the time of injection, five patients were classified as AIS grade A, two as AIS B, and three as AIS C. Patient 9 was injected at 22 months post-injury due to failure of adipose tissue expansion. A second biopsy was then re-obtained and expanded, resulting in an injection delay. The baseline characteristics of the enrolled patients are highlighted in Table 1.

### Safety and adverse events

In total, 44 AEs were reported among the ten patients, 17 of which (37%) were considered possibly related to the study drug. Of these 44 AEs, 26 were considered early AEs while 16 were considered late AEs. Eight of ten patients experienced headaches in the early follow-up period, while nine of ten patients experienced musculoskeletal symptoms, including back pain (two patients), general hip pain (three patients), tailbone pain or soreness (three patients), and leg pain (one patient). Over-the-counter medications were used in six of ten patients, which completely resolved their symptoms. No serious AEs were observed during the study period. Table 2 depicts the full safety profile for the ten patients. No abnormalities were found in the post-injection blood and CSF tests. No significant changes were seen on MRIs at follow-up; however, mildly increased clumping of the cauda equinia nerve roots with or without evidence of nodular enhancement was detected in eight of ten patients. No apparent clinical correlation was identified in the patients who experienced clumping or enhancement of the cauda equina nerve roots. Figure 1 shows the pre-infusion and follow-up MRI of patient 6, who developed mildly increased clumping of the cauda equina nerve roots at the L4-S1 level. Table 3 highlights the MRI changes for all patients before and after stem cell injection. Improvements in SSEPs were noted in three patients. Two patients did not have any change from baseline, an interpretation was not possible for four patients, and one patient had normal SSEPs throughout the study period. The SSEP findings and interpretations are highlighted in Table 4. Overall, the SSEP findings,

**Table 1 | Demographics and injury characteristics of all patients**

| Patient # | Injury Level | Race | AIS at Injury | AIS at Enrollment | AIS at Injection | AIS at Week 96 | Component driving AIS change after injection | Time between injury and enrollment | Time between injury and injection |
|---|---|---|---|---|---|---|---|---|---|
| Patient 1 | Cervical C5-C6 | White | A | C | C | D | More than half of the key muscles below the level of injury graded 3 or better | 10 months | 11 months |
| Patient 2 | Thoracic T11-T12 | White | A | A | A | A | | 10 months | 11 months |
| Patient 3 | Cervical C8 | White | B | B | B | C | Voluntary anal contraction | 8 months | 10 months |
| Patient 4 | Thoracic T12 | Hispanic/ Latino | A | B | C | C | | 10 months | 14 months |
| Patient 5 | Thoracic T10 | White | A | A | A | C | Deep anal pressure and voluntary anal contraction | 6 months | 8 months |
| Patient 6 | Cervical C6 | White | A | A | A | B | Deep anal pressure | 11 months | 13 months |
| Patient 7 | Thoracic T11 | Black | A | A | A | C | Deep anal pressure and motor function more than three levels below the motor level | 9 months | 10 months |
| Patient 8 | Cervical C4 | Asian | A | A | A | A | | 2 months | 7 months |
| Patient 9 | Cervical C7 | White | B | C | C | D | More than half of the key muscles below the level of injury graded 3 or better | 10 months | 22 months |
| Patient 10 | Cervical C5-C6 | White | A | B | B | C | Voluntary anal contraction | 12 months | 14 months |

**Table 2 | Primary evaluation of the adverse events for all patients**

| Patient # | Early AEs | Comments | Late AEs | Comments |
|---|---|---|---|---|
| Patient 1 | Headache, tailbone pain, blurry vision | Resolved with OTC | Heel pain, numbness of hands & feet, foot skin lesion, fatigue, spasticity, neuropathic pain in the chest | Resolved |
| Patient 2 | Headache, cold flu-like congestion, fever, low back pain | Resolved with OTC | Skin breakdown of heal, depression | Resolved |
| Patient 3 | Headache, tailbone pain, stomach pain | Resolved with OTC | Skin breakdown of heal, depression | Resolved |
| Patient 4 | Headache, general hip pain | Resolved | N/A | Resolved |
| Patient 5 | Sharp back pain, dizziness, hip pain | Resolved with OTC | Warming sensation felt in MRI, left buttock lesion, left hip pain | Resolved with Tramadol |
| Patient 6 | Foot pain | Resolved | Hip fracture (fall from the chair) | Not related to study |
| Patient 7 | Headache, bilateral hip and leg pain | Resolved | Sweating below the waist in both legs | Resolved |
| Patient 8 | Low-grade fever, arm pain, headache | Resolved with OTC | Bilateral lower leg spasms | Resolved |
| Patient 9 | Headache, tailbone soreness, tightness | Resolved | Skin breakdown on the ulnar aspect of the left fifth MCP | Resolved |
| Patient 10 | Headache and dysreflexia | Resolved with OTC | Upper back pressure injury | Resolved |

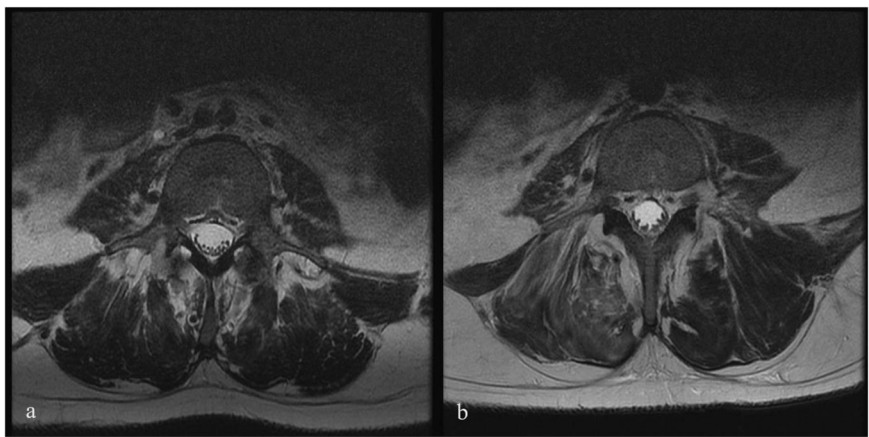

**Fig. 1 | Pre- and Post-Infusion MRI of Patient 6.** Axial view of T2-weighted Magnetic Resonance Imaging (MRI) of the lumbar spine of Patient 6, who developed cauda equina clamping following infusion at L4-S1. **a** Pre-infusion MRI scan at the L4 level. **b** Post-infusion MRI scan at the L4 level at 1-year follow-up.

when there was sufficient data for interpretation, correlated with the patients' clinical assessment. In that regard, patients 3 and 4 are worth of particular note. Patient 4, who was AIS grade A at the time of injury, displayed normal SSEPs throughout the study period. The baseline SSEP was performed just prior to the time of injection, when patient 4 was AIS grade C at the time of assessment, which is in keeping with the normal SSEP findings. Regarding patient 3, according to their baseline clinical examination, light touch was not affected as much as pinprick. In addition, this patient's AIS grade B is driven by the complete loss of motor function below the level of injury. Taking into account that SSEP mainly evaluates dorsal column function, which mediates light touch sensation (whereas pin prick is transferred predominantly through the spinothalamic tracts)[17,18], the SSEP findings are in keeping with the ISNCSCI examination and the changes seen at follow-up.

### AIS Grading

Each patient received ten physical examinations throughout the study period. In our cohort, seven of ten patients demonstrated an improvement in their AIS grades from the time of injection. Changes in the AIS grade at the time of injury, enrollment, injection, and follow-up are highlighted in Fig. 2. Five patients were classified as AIS A at the time of injection. Of those, two remained AIS A at the last follow-up, while three changed from AIS A to B (two patients) and to AIS C (one patient) at 96. Four of the five patients that were classified as AIS B or C at injection demonstrated improvement in AIS grade. The basis for the AIS changes for each patient is

included in Table 1. In addition, seven patients displayed improvement in motor and sensory function in at least one level, while two patients experienced improvement in sensation in at least one level. Supplementary Table 1 represents the ISNCSCI score changes from baseline for Motor, Sensory – Pin Prick, and Sensory – Light Touch at weeks 4, 24, 48, and 96. Supplementary Table 2 shows the number of ISNCSCI muscle function levels and dermatomes improved per patient at the final follow-up (compared to baseline) for Motor, Sensory – Pin Prick, and Sensory – Light Touch. Supplementary Figs. 1–10 represent the dermatomal body maps at baseline and final follow-up.

### CSF Parameters

Lumbar punctures were performed at baseline and two weeks post-injection. The median for CSF protein was 45 mg/dL at baseline (IQR = 56) and 46.5 mg/dL at follow-up (IQR = 38.5). Cerebrospinal fluid nucleated cells had a median of 2 cells/uL at baseline (IQR = 3) and 9.5 cells/uL at two weeks post-injection (IQR = 30.25). There was a decrease in CSF glucose from a median of 52 mg/dL at baseline (IQR = 4) to a median of 46.5 mg/dL (IQR = 8.5). Using the Cytokine Panel 1, GM-CSF, IL-16,17,1 A, 5,7, and TNF-β were below the limits of detection in CSF before and after AD-MSC injection. In this cohort, 6 of 10 patients had an undetectable VEGF level (defined as VEGF < 15.4 pg/ml) that later increased to a detectable threshold post-injection. Overall, seven of nine patients with available measurements demonstrated an increase in VEGF levels after AD-MSC injection. These changes are depicted in Fig. 3.

**Table 3 | MRI changes at baseline and one year after injection**

| Patient # | MRI pre-injection | MRI at one year | Comments |
|---|---|---|---|
| Patient 1 | Focal injury to the cervical cord at C3 with diffuse increased T2 signal in the cervical and thoracic cord below this level. | No significant change from the prior examination | N/A |
| Patient 2 | Myelomalacia of the thoracic spinal cord at the T11 and T12 levels | No significant change from the prior examination | N/A |
| Patient 3 | Focal T2 hyperintensity within the cervical cord at the C6-7 level consistent with posttraumatic myelomalacia. | No significant change from the prior examination | Mildly increased clumping of the cauda equina nerve roots at the L4-L5 level. |
| Patient 4 | Central cord signal abnormality that extends from T11 to the conus medullaris at T12-L1. More equivocal central T2 signal abnormality also extends cranially up to T6-T7. | No significant change from the prior examination | Clumping/thickening of the lumbar nerve roots. |
| Patient 5 | Cystic myelomalacia in the lower thoracic cord. | No significant change from the prior examination | Clumping of the roots of the cauda equina without evidence of nodular enhancement |
| Patient 6 | Cord signal abnormality consistent with myelomalacia that extends from C4-5 to C7, with subtle cord signal abnormality on the Sag IR sequence extending inferiorly to T2. | No significant change from the prior examination | Development of thickening, clumping, and minimal enhancement of the nerve roots of the cauda equina. |
| Patient 7 | Mild T2 signal abnormality in the dorsal cord, presumably related to Wallerian degeneration. | No significant change from the prior examination | New nodularity along the cauda equina at L2, L3, and in the cul-de-sac. Evidence of new prominence of the lower nerve roots. |
| Patient 8 | There is severe myelomalacia of the cervical spinal cord extending from the C5 through mid-C7. There is an irregular intrathecal enhancement that is involving the spinal cord, most prominently at the C6 level. | No significant change from the prior examination | Interval increase in the segmental enhancement of cauda equina nerve roots as well as leptomeningeal enhancement of the distal cord, extending cephalad to at least approximately mid-thoracic levels. |
| Patient 9 | Non-enhancing T2 hyperintense signal change within the spinal cord at the level of the C5-C6 interspace consistent with sequela of prior trauma. | No significant change from the prior examination | New clumping and peripheral displacement of the cauda equina nerve roots with mild enhancement. |
| Patient 10 | Marked myelomalacia of the cervical spinal cord extending from the lower body of C4 to the midbody of C7 with marked ventral kinking of the cord at the level of C5. | No significant change from the prior examination | Signal abnormality and clumping along with the cauda equina nerve roots at L3. |

## Discussion

This Phase I clinical trial demonstrates the safety of intrathecal administration of culture-expanded $1 \times 10^8$ cells of autologous AD-MSCs for patients with traumatic SCI. Stem cells were successfully manufactured, and products were delivered to all enrolled patients. No serious adverse events occurred throughout the study period, although non-serious adverse events were not infrequent. No patient initially intended to receive treatment was eventually excluded from the study. Several patients demonstrated sensory and motor improvement based on AIS impairment grade assessments.

Currently, there is a substantial body of evidence generated from preclinical studies indicating MSCs may potentially modulate various pathways involved in endogenous neurogenesis, angiogenesis, immunological regulation, and neuronal plasticity[19,20]. Compared with other adult stem cells, AD-MSCs are particularly advantageous, given their angiogenic and neuro-regenerative capacity, in addition to their pluripotency, ease of harvest, and availability[6]. In the present study, while non-serious AEs did occur, no serious AEs were documented during and after the AD-MSCs injection. These results are consistent with an earlier human trial investigating the safety of intrathecal injection of AD-MSCs in patients with SCI, where no serious AEs occurred among 14 treated patients. In that study, five patients exhibited upper or lower extremity motor improvements[21]. The safety profile has also been corroborated in a prior study with amyotrophic lateral sclerosis (ALS) conducted by Staff et al. 2016. In contrast to our study, Staff et al. involved a dose-escalation protocol ranging from $1 \times 10^7$ (single dose) to $1 \times 10^8$ cells (2 monthly doses). Nonetheless, intrathecal treatment of autologous AD-MSCs was also deemed safe[22]. Similar findings have also been established in other studies related to ALS, multiple sclerosis, and other neurodegenerative conditions[23,24].

In the present study, a significant proportion of patients exhibited varying degrees of cauda equina thickening, clumping, or nodular enhancement. The existing body of literature presents conflicting findings regarding the association of these imaging characteristics following intrathecal administration of stem cells with neurological deterioration, with some studies suggesting a link, while others failing to establish such a correlation. Notably, case reports have postulated that underlying inflammatory processes triggered by intrathecal stem cell administration may result in nerve root compression and subsequent symptom development[25–27]. Conversely, Singer et al. in their phase I/II trial employed a dose escalation design to assess the intrathecal administration of AD-MSCs in multiple system atrophy and revealed similar MRI findings in all patients within the medium (2 doses of $5 \times 10^7$ cells) and high-dose groups (2 doses of $1 \times 10^8$ cells)[28]. Their proposition centers on these MRI changes being reflective of a reactive response to the stem cell infusion, with more severe instances resembling features of reactive arachnoiditis. Importantly, their patients did not exhibit significant neurological deficits, albeit half of them experienced low-back discomfort. Likewise, the MRI changes observed in our patient cohort did not appear to correlate with neurological manifestations, thereby supporting a more benign interpretation of these findings. This intriguing observation remains an active area of research, and future studies incorporating pathological correlates are imperative to provide a definitive resolution to this question.

In our study, several patients demonstrated sensory and motor improvement based on AIS impairment grade assessments. However, these results are to be interpreted with caution given the intrinsic limitations of Phase I clinical trials. Drawbacks of the AIS grading have also been previously recognized[29]. Late neurologic recovery following spinal cord injury, as described by Kirshblum et al. at one year, is

**Table 4 | Somatosensory evoked potentials for patients at baseline, 12 months, and 24 months after AD-MSC injection or right upper, left upper, right lower and left lower limbs at each evaluation**

| Patient # | Baseline SSEP | First SSEP after Injection | Final SSEP after Injection | Overall Interpretation |
|---|---|---|---|---|
| Patient 1 | UE SSEP: Bilateral absolute scalp latencies prolonged. All bilateral interpeak latencies prolonged. LE SSEP: Bilateral absolute scalp latencies prolonged. Bilateral lumbar-to-scalp and left cervical-to-scalp interpeak latencies prolonged. | Performed at 5 months: UE SSEP: Bilateral absolute scalp latencies prolonged. Bilateral cervical-to-scalp and clavicle-to-scalp interpeak latencies prolonged. LE SSEP: Bilateral absolute scalp latencies prolonged. Bilateral lumbar-to-scalp interpeak latencies prolonged. | Performed at 18 months: UE SSEP: Bilateral absolute scalp and left absolute cervical latencies prolonged. Bilateral clavicle-to-scalp and left clavicle-to-cervical interpeak latencies prolonged. LE SSEP: Left absolute scalp latency prolonged. Left lumbar-to-scalp interpeak latency prolonged. | Mild improvement mainly in the right lower extremity SSEPs. |
| Patient 2 | UE SSEP: Normal left median potentials and latencies. Right median SSEP not performed. LE SSEP: Absent left cervical and scalp potentials. Normal left absolute lumbar potential and latency. Right tibial SSEP not performed. | Performed at 9 months: UE SSEP: Not performed. LE SSEP: Absent left cervical and scalp potentials. Normal left absolute lumbar potential and latency. No response elicited on the right. | Performed at 19 months: UE SSEP: Not performed. LE SSEP: No reproducible peripheral or central potentials. | No reliable interpretation. |
| Patient 3 | UE SSEP: Bilateral clavicle-to-scalp and clavicle-to-cervical interpeak latencies prolonged. LE SSEP: Normal. | Performed at 5 months: UE SSEP: Normal. LE SSEP: Normal. | Performed at 17 months: UE SSEP: Normal. LE SSEP: Normal. | Improvement in UE SSEP from abnormal at baseline to normal at 5 months. |
| Patient 4 | UE SSEP: Normal. LE SSEP: Normal. | Performed at 5 months: UE SSEP: Normal. LE SSEP: Normal. | Performed at 15 months: UE SSEP: Not performed. LE SSEP: Normal. | Normal throughout. |
| Patient 5 | UE SSEP: Not performed. LE SSEP: Bilateral absolute scalp potentials mildly prolonged. Left lumbar-to-scalp interpeak latency borderline prolonged. | Performed at 5 months: UE SSEP: Not performed. LE SSEP: No reliable information obtained. | Performed at 17 months: UE SSEP: Not performed. LE SSEP: Normal bilateral absolute lumbar potentials. No other reliable information obtained. | No reliable interpretation. |
| Patient 6 | UE SSEP: Right clavicle-to-cervical and left clavicle-to-scalp interpeak latency mildly prolonged. LE SSEP: Normal bilateral lumbar potentials. Normal left cervical absolute and lumbar-to-cervical interpeak latency. No other reliable information obtained. | Performed at 5 months: UE SSEP: Left clavicle-to-scalp interpeak latency borderline prolonged. Normal right absolute and interpeak latencies. LE SSEP: Normal left cervical and bilateral lumbar absolute latencies. Normal lumbar-to-cervical interpeak latency. No other reliable information obtained. | Performed at 17 months: UE SSEP: Left clavicle-to-scalp interpeak latency borderline prolonged. Normal right absolute and interpeak latencies. LE SSEP: Normal bilateral lumbar potentials. No other reliable information obtained. | No significant change from baseline. |
| Patient 7 | UE SSEP: Normal. LE SSEP: No reliable information obtained. | Performed at 7 months: UE SSEP: Not performed. LE SSEP: Normal bilateral lumbar absolute latencies. No other reliable information obtained. | Performed at 17 months: UE SSEP: Not performed. LE SSEP: Normal bilateral absolute lumbar latencies. Right absolute scalp and lumbar-to-scalp interpeak latencies prolonged. | No reliable interpretation. |
| Patient 8 | UE SSEP: Left cervical and scalp absolute latencies prolonged. All left interpeak latencies prolonged. Right clavicle-to-scalp interpeak latency prolonged. LE SSEP: No reliable information obtained. | Performed at 11 months: UE SSEP: Bilateral clavicle-to-cervical and clavicle-to-scalp interpeak latencies prolonged. LE SSEP: No reliable information obtained. | Performed at 16 months: UE SSEP: Bilateral clavicle-to-scalp interpeak latencies prolonged. Right clavicle-to-cervical interpeak latency prolonged. LE SSEP: Normal left cervical and right lumbar absolute latencies. No other reliable information obtained. | No change from baseline. |
| Patient 9 | UE SSEP: Bilateral clavicle-to-scalp and right clavicle-to-cervical interpeak latencies prolonged. LE SSEP: Bilateral lumbar-to-scalp interpeak latency prolonged. | Performed at 7 months: UE SSEP: Normal. LE SSEP: Left absolute scalp latency normal. No other reliable information obtained. | Performed at 15 months: UE SSEP: Normal. LE SSEP: Normal bilateral absolute scalp latencies. No other reliable information obtained. | Resolution of upper extremity central potential prolongation. No reliable interpretation for lower extremities. |
| Patient 10 | UE SSEP: Normal right absolute clavicle latency. No other reliable information obtained. LE SSEP: No reliable information obtained. | NA | NA | No reliable interpretation. |

Source data are provided as a Source Data file.
All upper extremity studies were performed via stimulation of the median nerve at the wrist, and all lower extremity studies were performed via stimulation of the tibial nerve at ankle.
*UE SSEP* upper extremity somatosensory evoked potential with median nerve (wrist) stimulation, *LE SSEP* lower extremity somatosensory evoked potential with tibial nerve (ankle) stimulation, *NA* not available

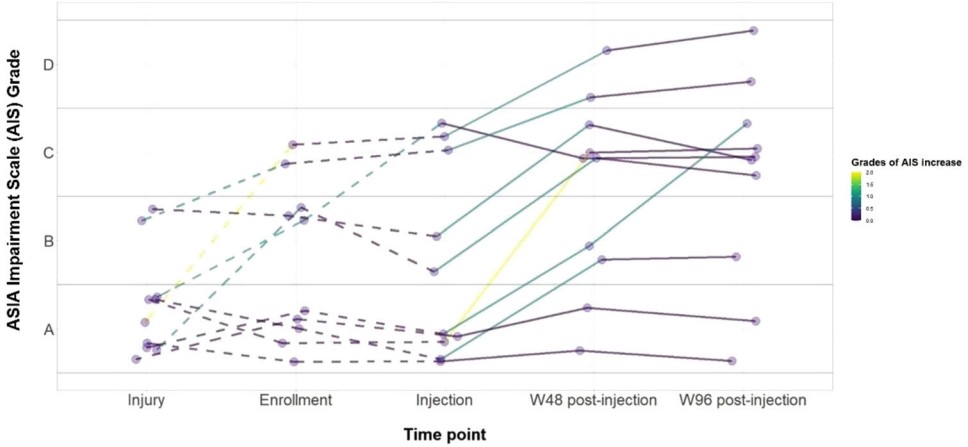

**Fig. 2 | ASIA Impairment Scale Grades.** AIS Grades at the time of injury, enrollment, injection, and final follow-up.

expected at a rate of approximately 5% for patients with an AIS grade A at baseline. In their longitudinal study, 3.5% of patients improved from AIS grade A to B at one year, and 1.05% improved from AIS grade A to C or D[30]. In our study, two of five patients (40%) improved from an AIS grade A at injection to C at final follow-up, demonstrating improvement following neurological plateau. However, the absence of controls prevents ascribing the observed neurological improvement solely to the administration of AD-MSCs. Furthermore, it's essential to consider that the patients included in the study were treated at a quaternary academic center with a highly specialized multidisciplinary team, which likely provided rehabilitation support that differs from what most spinal cord injury patients typically receive. As a result, there is a need for future, larger randomized controlled trials employing various clinimetric tools and quality of life measures, to investigate the potential benefits of AD-MSC injections in late-stage neurological recovery for SCI.

Preclinical and clinical studies suggest that MSC preservation in vivo is limited, and their favorable effects may be due to their ability to regulate tissue homeostasis and inflammation through the synthesis of various paracrine factors[31]. In this study, we observed an increase in the level of VEGF. This supports the proposed paracrine mechanism of action of MSCs in SCI; however, further investigation is needed to explore this association and potential mediating process. Furthermore, a variety of cytokines, such as IL-6, IL-10, and TNF-α, which were not included in the cytokine panel utilized in our study, have been implicated in the early and late post-injury phases[32–34]. Thus, it would be worthwhile for future studies to expand the assessment of cytokines and encompass a more extensive array of immunomodulatory and angiogenic markers.

This Phase I clinical trial (CELLTOP) investigated the safety profile of intrathecal AD-MSCs injection in ten patients with traumatic SCI. Overall, no serious adverse events were reported throughout the study period. At the final follow-up, seven of the ten patients experienced an improvement in AIS grade compared to their pre-injection status. The present safety profile of AD-MSC injection warrants further investigation with regard to the impact on the patients' neurological outcomes.

## Methods
### Standard protocol approvals – Registration – Patient consent
This Phase I single-arm, prospective, open-label study evaluating the use of intrathecal autologous AD-MSC therapy for patients with SCI, was reviewed and allowed to proceed by the United States Food and Drug Administration (ClinicalTrials.gov Identifier: NCT03308565) and the Mayo Clinic Institutional Review Board (IRB no. 17-004621). The clinical trial started in June 2017 and was completed in October 2021. The first patient was enrolled in November 2017 and the last patient

was enrolled in August 2019. Written informed consent was obtained from each participant prior to study enrollment. Data from one of the participants included in this study have been previously published as a case report[16]. Participants did not receive compensation for their participation.

### Selection criteria
Patients were considered eligible if they were 18 years or older and had a traumatic, blunt, non-penetrating, and non-degenerative injury within 12 months prior to study enrollment. Patients with an AIS grade of A or B at the time of the injury and with or without subsequent improvement within 12 months of injury were considered. The study design did not incorporate considerations of sex and gender, as variations in sex or gender were not expected to impact the safety of intrathecal administration of stem cells. To be enrolled, included patients had to fully understand the study requirements and comply with the treatment plan. The full selection criteria are available on ClinicalTrials.gov (Identifier: NCT03308565).

### Study design
A total of 14 patients with traumatic SCI were screened, and ten patients were enrolled and injected with AD-MSCs (Fig. 4). The reasons for selection failure for the four patients include non-traumatic etiology of injury, not meeting the inclusion criteria, and failure to consent. Patients first underwent fat harvest and subsequent AD-MSC isolation and expansion. These stem cells were expanded to a dosage of 100 million cells and injected at the lumbar level under fluoroscopic guidance. The dosage protocol employed in the present study was informed by a precedent Phase I dose-escalation trial, which studied the safety of intrathecal AD-MSC injections in amyotrophic lateral sclerosis (ALS)[22]. The reference trial utilized the same cellular product as the one under evaluation in our current study. It revealed a positive safety profile at the highest dose of a single dose of 100 million cells. The present study was carried out as a single-dose Phase I trial where all ten patients were closely followed for the occurrence of any adverse events (AEs). All patients were evaluated at the screening (pre-injection) visit, the baseline (injection) visit, and at all follow-up visits. Follow-up visits were scheduled on Day 2, Day 3, Week 1, Week 2, Week 6, Week 12, Week 24, Week 48, and Week 96 (Fig. 5). All participants were followed for two years after injection.

### AD-MSC collection, preparation, and administration
After the patient was considered eligible for study inclusion and consented to participate, enrollment into the study commenced with a scheduled fat harvest. The patient underwent a fat biopsy through a small (1–2 inch) surgical incision in the abdomen or thigh, where

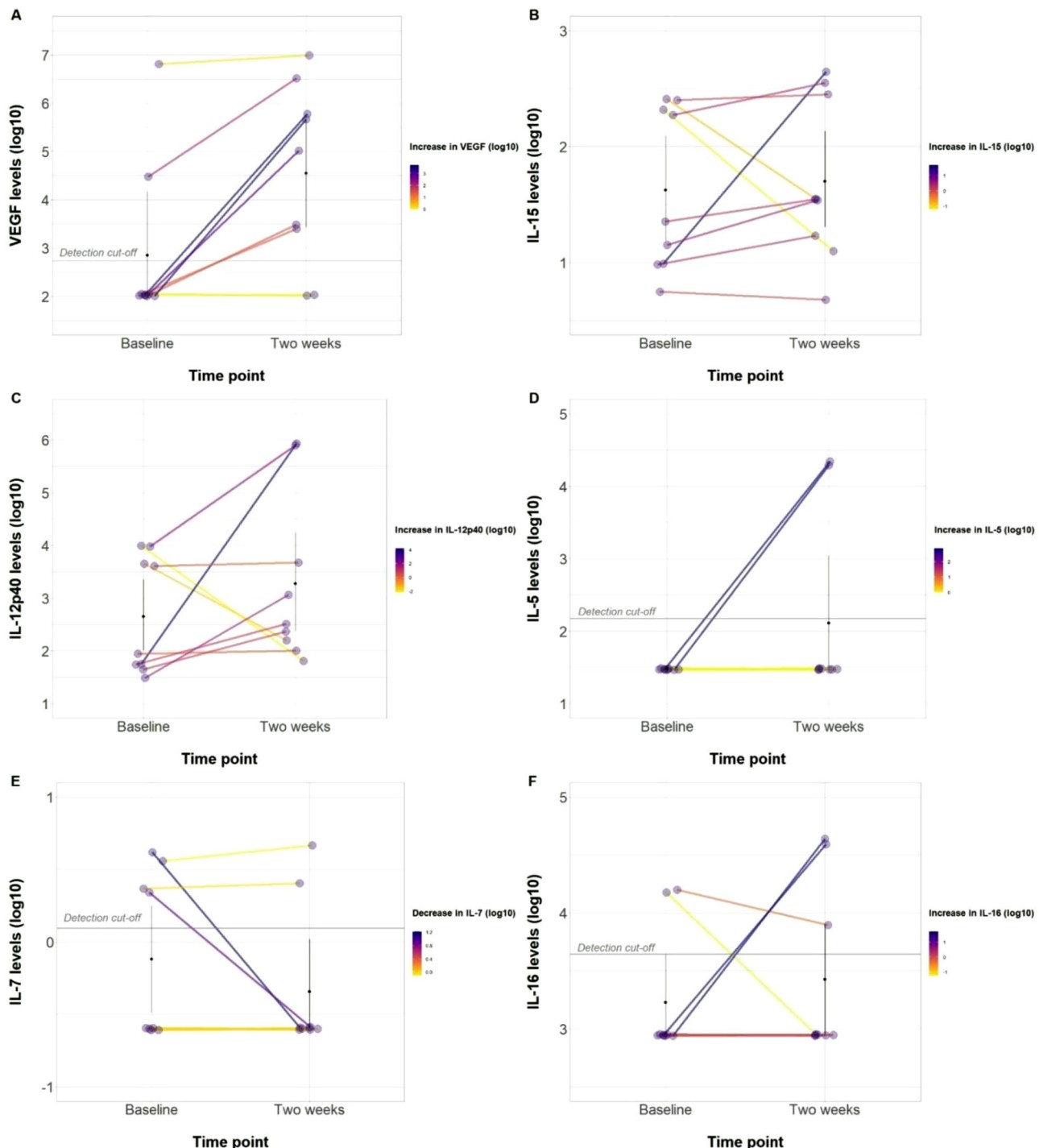

**Fig. 3 | Cerebrospinal Fluid Cytokine Changes.** Changes in Cerebrospinal Fluid (CSF) parameters after AD-MSCs injection. **A** VEGF levels before and 2-weeks after infusion. **B** IL-15 levels before and 2-weeks after infusion. **C** IL12p40 levels before and 2-weeks after infusion. **D** IL-5 levels before and 2-weeks after infusion. **E** IL−7 levels before and 2-weeks after infusion. **F** IL-16 levels before and 2-weeks after infusion. Source data are provided as a Source Data file.

approximately 15 mL of adipose tissue was removed from underneath the skin. This procedure was performed by a certified registered nurse. The patient was administered a local anesthetic prior to the incision, and the wound was closed with sutures following the harvest. The extracted adipose tissue was processed for AD-MSC isolation and culture expansion to 100 million cells, as previously described in the literature[28]. The following description of the cell therapy product is based on the DOSES guidelines[35]. The cell-manufactured product was an autologous derived from adipose tissue and purified by culture flask adherence. Cells were expanded in 5% platelet lysate-based media (PLTMax; Mill Creek LifeSciences, Rochester, MN). Cells were expanded until sufficient cell dosage was achieved and cryopreserved for release testing. Release testing consisted of sterility tests (mycoplasma, aerobic and anaerobic culture growth), genetic stability (karyotype analysis), and purity and potency assays using flow cytometry as recommended[36]. Isolation, expansion, and release testing was completed 4–6 weeks after AD-MSC isolation from the adipose tissues. Upon release, the cells were thawed and allowed to recover for up to

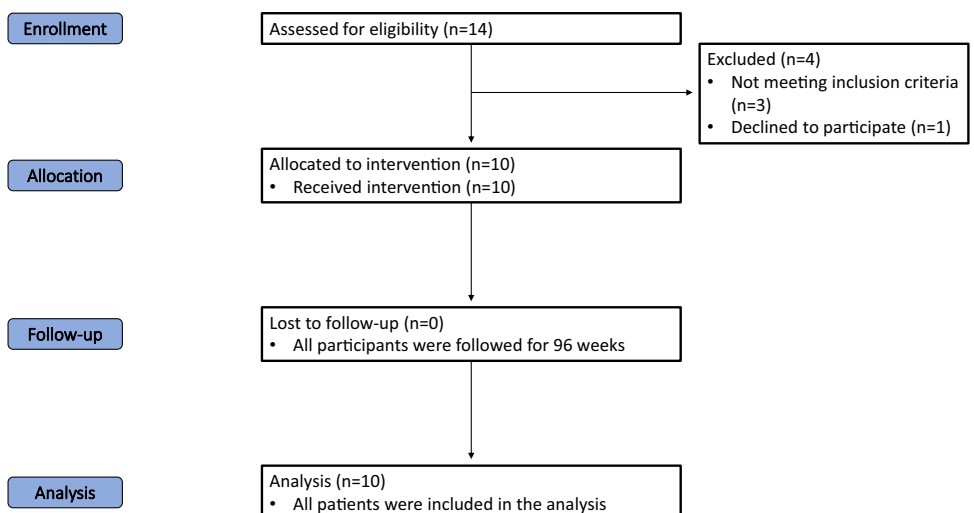

**Fig. 4 | CONSORT diagram.** Flow diagram of the progress through the phases of the trial (enrolment, allocation, follow-up, and analysis).

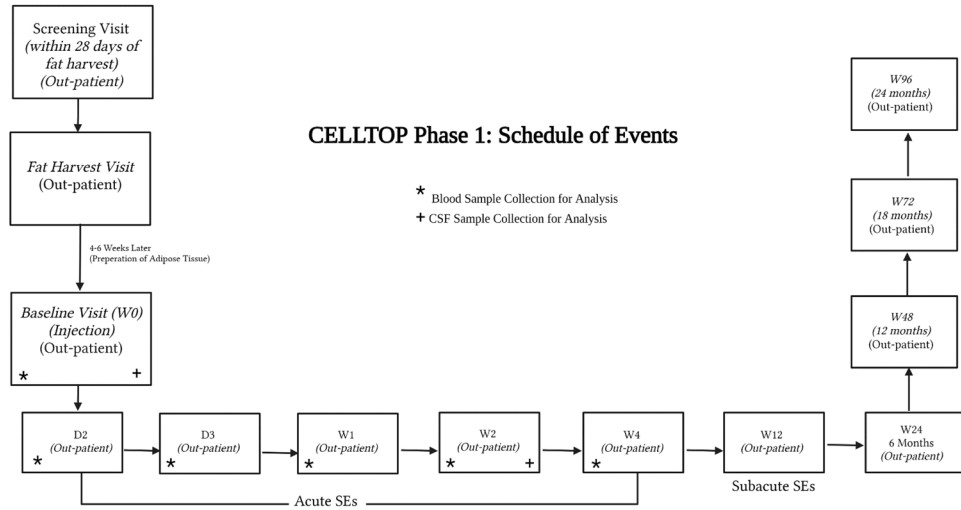

**Fig. 5 | Study Design.** The study protocol and scheduled patient visits.

four days in media. On the day of treatment, cells were collected, washed, and resuspended in lactated Ringer's solution for intrathecal delivery at the designated dose of 100 million cells in 10 ml.

After completing the baseline assessments, all participants were admitted to Mayo Clinic's Inpatient Clinical Research and Trials Unit the day before injection and remained hospitalized for 24 h after the injection. On the day of the procedure, the product was delivered in a controlled container at 2 °C–8 °C. A lumbar spinal needle was placed in the subarachnoid space via a standard posterior intervertebral approach between lumbar levels 2 and 5; each patient's specific level was determined individually based on anatomical considerations. After the collection of CSF samples for baseline analysis, MSCs were infused into the CSF over 2–3 min via free-hand delivery by one of the study physicians, followed by a 1 mL flush with lactated Ringer's solution. After cell injection, the patient was rotated side-to-back-to-side every 15 min for 2 h to maximize the even distribution of cells within the CSF. The process of fat harvest, AD-MSCs isolation, culture, and administration is illustrated in Fig. 6.

## Assessment

The primary endpoint of this study was the safety profile of AD-MSCs, as reflected via the nature, incidence, and severity of any AEs. Adverse events were defined as any untoward or undesirable medical occurrence in the form of signs, symptoms, abnormal findings, or diseases that emerge or worsen relative to baseline (regardless of whether the AE had a causal relationship with the study drug). Serious AEs were defined as events that involve any of the following: death, life-threatening adverse experience, new inpatient hospitalization or prolonged hospitalization, disability, or birth defect/anomaly. Blood samples were collected at week 1 and 4 post-injection, and complete blood count and basic biochemical panel was assessed for safety surveillance. Early AEs were defined as AEs that occurred two weeks after injection, and late AEs were defined as AEs that occurred after two weeks. Adverse events were obtained through spontaneous subject reports, subject interviews by study personnel, medical chart review, clinical examinations during follow-ups, and imaging.

The secondary endpoints for this study were changes in patients' sensory and motor scores. Neurologic level of injury and severity scores were assessed by Physical Medicine and Rehabilitation Physicians or Advanced Nurse Practitioners, which included AIS grading. Each patient was scheduled to receive a total of ten examinations after injection. The AIS grading was used to assess the sensory and motor levels of each patient. The scale has five classification levels, ranging from complete loss of neural function in the affected area (Grade A) to completely normal (Grade E)[37]. Magnetic resistance images of the spine, somatosensory evoked potentials, and CSF cytokines analyses were performed.

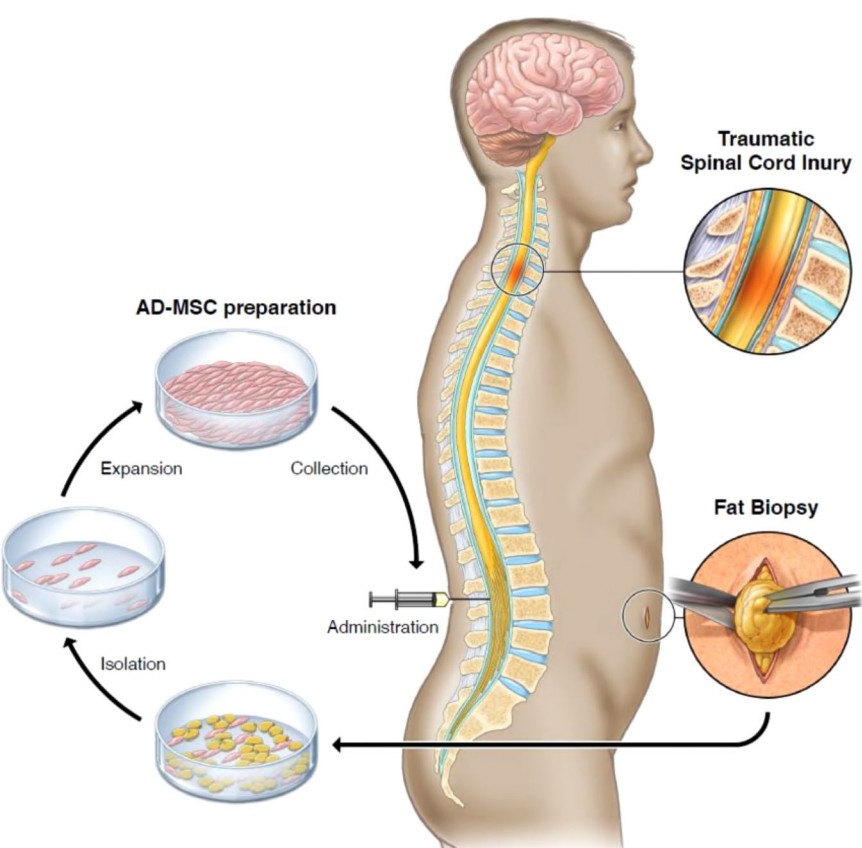

**Fig. 6 | Fat harvest via biopsy, AD-MSCs isolation, expansion, and collection, and treatment administration.** Illustration of the process of fat harvest via biopsy, AD-MSCs isolation, expansion, and collection, and treatment administration.

### CSF Cytokine assay

Cerebrospinal fluid was collected prior to cell administration and two weeks post-injection and was immediately centrifuged at 1500 g for 10 min, aliquoted, and stored at −80 °C. Post-injection samples were analyzed for cell count and a basic biochemical panel for safety surveillance. Nine sample pairs of treated participants with available baseline and two weeks post-injection samples were analyzed using a Multiplex immunoassay (V-PLEX Cytokine Panel 1 Human Kit, Meso Scale Diagnostics, Rockville, MD, USA) according to the manufacturer's instructions. We were unable to collect the CSF from patient 6 due to discomfort during the lumbar puncture. The Cytokine Panel 1 (human) Kit provided assay-specific components for the quantitative determination of GM-CSF, IL-1α, IL-5, IL-7, IL-12/IL-23p40, IL-15, IL-16, IL-17A, TNF-β, and VEGF-A. The cytokine values in the CSF were transformed into log10 to present the change after AD-MSC injection.

### Statistics & reproducibility

No statistical method was used to predetermine sample size. The sample size was determined based on clinical considerations rather than statistical analysis. No data were excluded from the analyses. As a single arm Phase I trial, the participants were not randomized and the investigators were not blinded to allocation during experiments and outcome assessment.

### Reporting summary

Further information on research design is available in the Nature Portfolio Reporting Summary linked to this article.

### Data availability

The somatosensory evoked potentials (Table 4) and the cerebrospinal fluid cytokine levels (Fig. 3) are provided in the source data file with this paper. The processed data from the patients' electronic health records, pertaining to adverse events, imaging changes, and sensory/motor changes are presented in Tables 1–3, Supplementary Tables 1, 2, Supplementary Images 1–10. For ethical and legal considerations pertaining to patient confidentiality, individual participant data relevant to the results presented in this article, will only be available upon request and after deidentification to protect patient identities. The data will be shared with investigators who provide a methodologically sound proposal and/or have obtained approval by an independent review committee. Shared data can only be used for the purposes of the approved proposal and/or individual participant data meta-analysis and will be available indefinitely. Data requests will be managed by the corresponding author via email (bydon.moha-mad@mayo.edu). The study protocol is available on ClinicalTrials.gov (Identifier: NCT03308565). Source data are provided with this paper.

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

## Acknowledgements

This research was made possible with support from Abdulmohsin Almishari, Hiba Almishari, C and A Johnson Family Foundation, Leonard A. Lauder, The Park Foundation, Sanger Family Foundation, Eileen R.B. and Steve D. Scheel, Schultz Family Foundation and other generous Mayo Clinic benefactors. We would also like to acknowledge Ms. Julia Thebiay and the Department of Laboratory Medicine and Pathology at Mayo Clinic for their support for this project. The corresponding author receives financial support as Charles B. and Ann L. Johnson Professor of Neurosurgery.

## Author contributions

M.B. was the principal investigator and conceptualized and supervised the study throughout all stages. W.Q contributed to the study plan and supervision. F.M.M. contributed to the study design and was involved in the cerebrospinal fluid cytokine analysis. A.J.W. was involved in the design and supervision of the study. K.L.G., C.L.H., R.K.R, K.D.Z., K.L.S.M., L.A.B., participated in the investigation and project administration of the study. B.C.T. assisted with the study administration. R.J. was involved in the study administration and manuscript writing. S.E.S., G.D.M., K.K. involved in the data analysis, manuscript writing, and visualization of the results. S.P.G. provided the illustrations included in the study. R.S.L. supervised the electrophysiological studies performed as part of the trial. A.B.D. was involved in the study design and the supervision of the stem cell processing.

## Competing interests

The authors declare no competing interests.

## Additional information

**Supplementary information** The online version contains Supplementary Material available at https://doi.org/10.1038/s41467-024-46259-y.

