## [Peer Review File · Nature Communications]

Intrathecal Delivery of Adipose-Derived Mesenchymal Stem Cells in Traumatic Spinal Cord Injury: Phase I TrialEditorial Note: Parts of this Peer Review File have been redacted as indicated to maintain patient confidentiality.

REVIEWER COMMENTS

Reviewer #1 (Remarks to the Author):

A Phase 1 study of intrathecal injection of fat-derived mesenchymal stem cells in initially motor complete subjects observed for 2 years. Cell manufacturing was successful. By week 7/10 subjects have AIS conversion. VEGF was increased in CSF after 2 weeks from below detection level to a detectable level on a log scale. Other cytokine changes were inconsistent. Figure 4 does not show the units for cytokine levels.

1) It is notable that nerve root clumping in the cauda equina was detected in all evaluated patients (8/10). This deserves more attention given the increase in VEGF. When were the follow-up MRIs performed? Examples of this should be shown with the pre-infusion and followup images. If this finding is not associated with any evident harm, that is important knowledge.

2) The ISNCSCI reporting is insufficient. Each subject's changes in motor and sensory scores need to be provided, and it would be a useful service to provide all data in a spreadsheet supplement.

3) The basis for changes in AIS in each subject need to be reported-? DAP for AIS B conversion,? VAC for AIS C conversions, and motor scores for each subject and especially subjects 1 and 9.

4) It's very difficult to know if these AIS changes indicate a therapeutic effect for several reasons, especially in the absence of concurrent controls. But there are few systematic studies tracking AIS change up to 2 years. Further, rehabilitation effort needs to be considered.

5) SSEP reporting is sloppy, and it would be beneficial to provide the actual data in a table or spreadsheet as there is so little data in the literature in cell therapy studies. For example, subject 4 is of interest as AIS B enrollment with a report of normal SSEPs. This correlation alone is notable and should be supported with the actual data and waveforms. Another example is patient 3 C6/7 AIS B to C reported to have a normal LE SSEP at baseline. This is unexpected and should be correlated to the ISNCSCI sensory exam.

6) Did changes in the sensory levels occur?

Reviewer #2 (Remarks to the Author):

This is a well-conducted Phase I trial on the Biological and Clinical Responses to Intrathecal Delivery of Autologous Adipose-Derived Mesenchymal Stem Cells in Patients with Traumatic Spinal Cord Injury. The study design, methods, and presentation (tables and figures) and interpretation of the results are mostly adequate.

However, there is one issue in trial design needing attention. A typical phase I design is 3+3 with dose escalation, but we didn't see this in this phase I trial. Why was the dosage of 100 million cells chosen? Why the trial is a single dosage trial and didn't escalate dosage as there is no SAE (toxicity)?

Reviewer #3 (Remarks to the Author):

The manuscript by Bydon et al. entitled “Biological and clinical responses to intrathecal delivery of autologous adipose-derived mesenchymal stem cells in patients with traumatic spinal cord injury: a phase 1 trial” describes the clinical outcomes in 10 spinal cord injury patients enrolled in a phase 1 clinical trial evaluating the intrathecal administration of autologous adipose tissue-derived mesenchymal stem cells. Of the ten enrolled patients, six had a cervical-level injury, and four had a thoracic-level injury. All were enrolled within 12 months of injury and had injuries classified as AIS A or AIS B.

The data presented is a look at a larger dataset from the CELLTOP trial, from which data on a single patient had been published previously. Overall, the manuscript presents noteworthy results found in the patients in the trial. Those described were that autologous AD-MSc intrathecal infusions were well tolerated with minimal adverse events. More notably, 7 of the ten patients had improvements in their AIS grade from the time of injection, with three demonstrating noticeable improvements in motor and sensory function. The report on this small number of patients could be important in the SCI treatment field, though similar trials in SCI have presented mixed results.

While the data looks very promising, the reviewer has several significant issues that reduce the enthusiasm for the manuscript as presented.

It is unclear whether the data on the previously published patient is part of the current dataset or are these ten distinct patients.

The Materials and Methods as presented are insufficiently detailed, so the work could not be reproduced as prepared.

The section on the isolation and preparation of AD-MSCs is woefully inadequate. They refer to a previous publication (Singer et al., 2019) to address cell preparation. However, the previous publication has no information on cell isolation, characterization and testing. As prepared, the reader is not indicated what they were administering to patients.

Moreover, the cell dosing is not described at all. There is a cursory mention of cell dose in the Discussion. Later in the Discussion, the authors mention this being a dose escalation study; however, it is not described in detail, which is completely unacceptable. Which patients got a single dose versus multiple doses? Were the multiple-dose patients the ones who saw marked improvements? It is never discussed.

Also, several figures mentioned in the manuscript do not align with the actual figures in the paper. There does not appear to be Figure 1B or 2B, as mentioned, which causes the other figures to mislabel.

I also feel that the authors failed to give a thoughtful summary and discussion of their data.

My overall assessment is that this manuscript has been carelessly assembled, which causes the reviewer to have serious concerns about its science.

Reviewer #4 (Remarks to the Author):

This report is on the CELLTOP phase I study which included tetra and paraparetic spinal cord injured patients receiving adipose tissue-derived mesenchymal stem cells (AD-MSc) 8-22 months after SCI. This work represents a considerable effort and is one of the very few interventional studies in the field of spinal cord injury, using state of the art methodology techniques (incl SSEP). The results demonstrate safety and expectable transient AEs. The reported improvement may just represent spontaneous recovery (plus placebo-effect), being unrelated to intervention itself. Overall, I think this is relevant report, but after incorporation of the suggestions for revision, it should be more suitable for specialized journals (some examples are listed below).

Phase 1 Safety Trial of Autologous Human Schwann Cell Transplantation in Chronic Spinal Cord Injury. Gant KL, Guest JD, Palermo AE, Vedantam A, Jimshelishvili G, Bunge MB, Brooks AE, Anderson KD, Thomas CK, Santamaria AJ, Perez MA, Curiel R, Nash MS, Saraf-Lavi E, Pearse DD, Widerström-Noga E, Khan A, Dietrich WD, Levi AD. *J Neurotrauma*. 2022 Feb;39(3-4):285-299. doi: 10.1089/neu.2020.7590. Epub 2021 May 3.

Safety of Autologous Human Schwann Cell Transplantation in Subacute Thoracic Spinal Cord Injury. Anderson KD, Guest JD, Dietrich WD, Bartlett Bunge M, Curiel R, Dididze M, Green BA, Khan A, Pearse DD, Saraf-Lavi E, Widerström-Noga E, Wood P, Levi AD. *J Neurotrauma*. 2017 Nov 1;34(21):2950-2963. doi: 10.1089/neu.2016.4895. Epub 2017 Mar 21.

Emerging Safety of Intramedullary Transplantation of Human Neural Stem Cells in Chronic Cervical and Thoracic Spinal Cord Injury.

Levi AD, Okonkwo DO, Park P, Jenkins AL 3rd, Kurpad SN, Parr AM, Ganju A, Aarabi B, Kim D, Casha S, Fehlings MG, Harrop JS, Anderson KD, Gage A, Hsieh J, Huhn S, Curt A, Guzman R. *Neurosurgery*. 2018 Apr 1;82(4):562-575. doi: 10.1093/neuros/nyx250.

Minor

1. Abstract: More important than the timepoint when patient has been enrolled is when the patient has received the LP (receiving intervention).
2. Figure 3: Y-axis label in Figure 3 of the axis should be "AIS Grade" (not 'ASIA Scores').
3. AD-MSc were propagated to 100 Million cells. Was this also the amount of cells injected into the patient ? How large was the volume injected into the CSF?

REVIEWER COMMENTS

Reviewer #1 (Remarks to the Author):

A Phase 1 study of intrathecal injection of fat-derived mesenchymal stem cells in initially motor complete subjects observed for 2 years. Cell manufacturing was successful. By week 7/10 subjects have AIS conversion. VEGF was increased in CSF after 2 weeks from below detection level to a detectable level on a log scale. Other cytokine changes were inconsistent. Figure 4 does not show the units for cytokine levels.

1) It is notable that nerve root clumping in the cauda equina was detected in all evaluated patients (8/10). This deserves more attention given the increase in VEGF. When were the follow-up MRIs performed? Examples of this should be shown with the pre-infusion and follow-up images. If this finding is not associated with any evident harm, that is important knowledge.

Response to reviewer: We thank the reviewer for their comment. All patients underwent MRI at baseline and 1 year after injection. No clinical correlation was identified in the patients who experienced clumping, or enhancement of the cauda equina nerve roots post-infusion. As per the reviewer's recommendation, we have included the pre-infusion and the last follow-up MRI of patient 6 to better illustrate this finding. In addition, we have elaborated on this observation in further detail in the discussion.

Change to text: Lines 206-209: "No apparent clinical correlation was identified in the patients who experienced clumping or enhancement of the cauda equina nerve roots. Figure 3 shows the pre-infusion and follow-up MRI of patient 6, who developed mildly increased clumping of the cauda equina nerve roots at the L4-S1 level."

Change to text: Lines 271-286: "In the present study, a significant proportion of patients exhibited varying degrees of cauda equina thickening, clumping, or nodular enhancement. The existing body of literature presents conflicting findings regarding the association of these imaging characteristics following intrathecal administration of stem cells with neurological deterioration, with some studies suggesting a link, while others failing to establish such a correlation. Notably, case reports have postulated that underlying inflammatory processes triggered by intrathecal stem cell administration may result in nerve root compression and subsequent symptom development.²⁹⁻³¹ Conversely, Singer et al. in their phase I/II

trial employed a dose escalation design to assess the intrathecal administration of AD-MSCs in multiple system atrophy and revealed similar MRI findings in all patients within the medium (2 doses of 5×10^7 cells) and high-dose groups (2 doses of 1×10^8 cells).¹⁸ Their proposition centers on these MRI changes being reflective of a reactive response to the stem cell infusion, with more severe instances resembling features of reactive arachnoiditis. Importantly, their patients did not exhibit significant neurological deficits, albeit half of them experienced low-back discomfort. Likewise, the MRI changes observed in our patient cohort did not appear to correlate with neurological manifestations, thereby supporting a more benign interpretation of these findings. This intriguing observation remains an active area of research, and future studies incorporating pathological correlates are imperative to provide a definitive resolution to this question.”

Change to text: Lines 477-481: Figure 3. Axial view of T2-weighted Magnetic Resonance Imaging (MRI) of the lumbar spine of Patient 6, who developed cauda equina clamping following infusion at L4-S1. a Pre-infusion MRI scan at the L4 level. b Post-infusion MRI scan at the L4 level at 1-year follow-up.

2) The ISNCSCI reporting is insufficient. Each subject's changes in motor and sensory scores need to be provided, and it would be a useful service to provide all data in a spreadsheet supplement.

Response to reviewer: We kindly thank the reviewer for their insightful comment. To better present the ISNCSCI score changes, we have included two additional tables (Table 5 and Table 6). Table 5 represents the motor, sensory – pinprick, and sensory – light touch score changes from baseline at weeks 4, 24, 48, and 96. In addition, Table 6 illustrates the number of ISNCSCI levels improved per patient.

Change to text: Lines 231-236: "**Table 5** represents the ISNCSCI score changes for Motor, Sensory – Pin Prick, and Sensory – Light Touch from baseline at weeks 4, 24, 48, and 96. **Table 6** shows the number of ISNCSCI muscle function levels and dermatomes improved per patient at the final follow-up for Motor, Sensory – Pin Prick, and Sensory – Light Touch.

Lines 356-459: "**Table 5:** Average International Standards for Neurological Classification of Spinal Cord Injury (ISNCSCI) Motor, Sensory – Pin Prick, and Sensory – Light Touch score changes from baseline at weeks 4, 24, 48, and 96."

Patient #	Average Motor Change				Average Pin Prick Change				Average Light Touch Change			
	Week 4	Week 24	Week 48	Week 96	Week 4	Week 24	Week 48	Week 96	Week 4	Week 24	Week 48	Week 96
Patient 1	13	18	21	25	18	25	22	26	10	17	25	42
Patient 2	0	1	1	1	0	1	-4	0	1	1	-3	1
Patient 3	1	-5	-6	1	23	14	19	29	19	10	5	11
Patient 4	5	4	2	0	10	7	-9	-3	-1	0	-1	-6
Patient 5	-1	-3	-2	-3	-4	-2	-2	-4	-3	-2	-3	-4
Patient 6	-1	2	-1	1	9	6	7	6	8	5	2	3
Patient 7	0	0	0	2	0	-1	0	1	0	0	3	0
Patient 8	-1	-2	-1	-2	4	2	5	1	2	2	0	-3
Patient 9	27	24	34	26	1	-8	11	-11	-7	-9	-4	-13
Patient 10	4	8	11	12	5	-1	-6	3	4	-3	-6	7

Lines 463-466: "**Table 6:** Number of International Standards for Neurological Classification of Spinal Cord Injury (ISNCSCI) muscle function levels and dermatomes improved at the final follow-up for Motor, Sensory – Pin Prick, and Sensory – Light Touch."

Patient #	Motor Change Improvement					Pin Prick Improvement			Light Touch Improvement			
	Number of ISNSCI levels improved	ISNSCI levels improved by 1	ISNSCI levels improved by 2	ISNSCI levels improved by 3	ISNSCI levels improved by 4	ISNSCI levels improved by 5	Number of ISNSCI levels improved	ISNSCI levels improved by 1	ISNSCI levels improved by 2	Number of ISNSCI levels improved	ISNSCI levels improved by 1	ISNSCI levels improved by 2
Patient 1	19	13	6				24	22	2	40	38	2
Patient 2	1	1								1	1	
Patient 3	1	1					28	27	1	11	11	
Patient 4							3	3		1	1	
Patient 5												
Patient 6	1	1					5	1	4	5	4	1
Patient 7	2	2					1	1				
Patient 8							2	1	1	2	1	1
Patient 9	12	5	2	4		1	2	2		3	3	
Patient 10	8	5	2	1			2	1	1	7	7	

3) The basis for changes in AIS in each subject need to be reported-? DAP for AIS B conversion,? VAC for AIS C conversions, and motor scores for each subject and especially subjects 1 and 9.

Response to reviewer: We thank their reviewer for their insight. One patient improved from AIS A to B following the injection as he regained deep anal pressure (DAP), while another improved from AIS A to C as they regained DAP and voluntary anal contraction (VAC). Two patients improved from AIS B to C as they both experienced restoration of voluntary anal contraction. Two patients improved from AIS C to D as they exhibited motor function greater than 3 in more than half of the key muscles below the level of injury. This information is now included in Table 1 under the column "Reason for AIS change after injection."

Change to text: Line 231: The basis for the AIS changes for each patient is included in **Table 1**.

[REDACTED]

4) It's very difficult to know if these AIS changes indicate a therapeutic effect for several reasons, especially in the absence of concurrent controls. But there are few systematic studies tracking AIS change up to 2 years. Further, rehabilitation effort needs to be considered.

Response to reviewer: We thank the reviewer for their comment. Indeed, with a relatively small sample size and lack of controls, therapeutic effect cannot be readily attributed to the intervention. However, in light of this study being a phase I trial, it was designed to evaluate the safety profile of intrathecally delivered adipose-tissue derived mesenchymal stem cells. We have included this point in the discussion.

Change to text: Lines 296-301: “However, the absence of controls prevents ascribing the observed neurological improvement solely to the administration of AD-MSCs. Furthermore, it's essential to consider that the patients included in the study were treated at a quaternary academic center with a highly specialized multidisciplinary team, which likely provided rehabilitation support that differs from what most spinal cord injury patients typically receive. As a result, there is a need for future, larger double-blinded, randomized controlled trials to investigate the potential benefits of AD-MSC injections in late-stage neurological recovery for SCI.”

5) SSEP reporting is sloppy, and it would be beneficial to provide the actual data in a table or spreadsheet as there is so little data in the literature in cell therapy studies. For example, subject 4 is of interest as AIS B enrollment with a report of normal SSEPs. This correlation alone is notable and should be supported with the actual data and waveforms. Another example is patient 3 C6/7 AIS B to C reported to have a normal LE SSEP at baseline. This is unexpected and should be correlated to the ISNCSCI sensory exam.

Response to reviewer: We thank the reviewer for their comment. We reviewed all available SSEPs and modified Table 4 to increase the clarity and accuracy of the SSEP findings. Specifically, some SSEP studies that were not able to provide reliable information were previously marked as "not performed" in the table. This has been rectified, and the text has been modified accordingly to "no reliable information obtained". In addition, given that SSEP examinations were not conducted at fixed intervals, we renamed the columns to "First SSEP after injection" and "Last SSEP after injection". Overall, where adequate information was available, SSEPs align with the clinical examination. The reviewer raises two interesting cases regarding the SSEP findings. The baseline SSEP was performed just prior to the time of injection, therefore patient 4 was AIS grade C at the time of assessment, which is in keeping with the normal SSEP findings. Regarding patient 3, their AIS grade B is driven by the complete loss of motor function below the level of injury. The text has been modified to highlight these cases.

Change to text: Lines 211-223: “Two patients did not have any change from baseline, an interpretation was not possible for four patients, and one patient had normal SSEPs throughout the study

period. The SSEP findings and interpretations are highlighted in **Table 4**. Overall, the SSEP findings, when there was sufficient data for interpretation, correlated with the patients' clinical assessment. In that regard, patients 3 and 4 are worth of particular note. Patient 4, who was AIS grade A at the time of injury, displayed normal SSEPs throughout the study period. The baseline SSEP was performed just prior to the time of injection, when patient 4 was AIS grade C at the time of assessment, which is in keeping with the normal SSEP findings. Regarding patient 3, according to their baseline clinical examination, light touch was not affected as much as pinprick. In addition, this patient's AIS grade B is driven by the complete loss of motor function below the level of injury. Taking into account that SSEP mainly evaluates dorsal column function, which mediates light touch sensation (whereas pin prick is transferred predominantly through the spinothalamic tracts),^{22,23} the SSEP findings are in keeping with the ISNCSCI examination and the changes seen at follow-up.”

Change to text: Lines 450-455: **Table 4**. Somatosensory evoked potentials for patients at baseline, 12 months, and 24 months after AD-MSJ injection or right upper, left upper, right lower and left lower limbs at each evaluation.

Patient #	Baseline SSEP	First SSEP after Injection	Final SSEP after Injection	Overall Interpretation
Patient 1	UE SSEP: Bilateral absolute scalp latencies prolonged. All bilateral interpeak latencies prolonged. LE SSEP: Bilateral absolute scalp latencies prolonged. Bilateral lumbar-to-scalp and left cervical-to-scalp interpeak latencies prolonged.	Performed at 5 months: UE SSEP: Bilateral absolute scalp latencies prolonged. Bilateral cervical-to-scalp and clavicle-to-scalp interpeak latencies prolonged. LE SSEP: Bilateral absolute scalp latencies prolonged. Bilateral lumbar-to-scalp interpeak latencies prolonged.	Performed at 18 months: UE SSEP: Bilateral absolute scalp and left absolute cervical latencies prolonged. Bilateral clavicle-to-scalp and left clavicle-to-cervical interpeak latencies prolonged. LE SSEP: Left absolute scalp latency prolonged. Left lumbar-to-scalp interpeak latency prolonged.	Mild improvement mainly in the right lower extremity SSEPs.
Patient 2	UE SSEP: Normal left median potentials and latencies. Right median SSEP not performed. LE SSEP: Absent left cervical and scalp potentials. Normal left absolute lumbar potential and latency. Right tibial SSEP not performed.	Performed at 9 months: UE SSEP: Not performed. LE SSEP: Absent left cervical and scalp potentials. Normal left absolute lumbar potential and latency. No response elicited on the right.	Performed at 19 months: UE SSEP: Not performed. LE SSEP: No reproducible peripheral or central potentials.	No reliable interpretation.

Patient 3	UE SSEP: Bilateral clavicle-to-scalp and clavicle-to-cervical interpeak latencies prolonged. LE SSEP: Normal.	Performed at 5 months: UE SSEP: Normal. LE SSEP: Normal.	Performed at 17 months: UE SSEP: Normal. LE SSEP: Normal.	Improvement in UE SSEP from abnormal at baseline to normal at 5 months.
Patient 4	UE SSEP: Normal. LE SSEP: Normal.	Performed at 5 months: UE SSEP: Normal. LE SSEP: Normal.	Performed at 15 months: UE SSEP: Not performed. LE SSEP: Normal.	Normal throughout.
Patient 5	UE SSEP: Not performed. LE SSEP: Bilateral absolute scalp potentials mildly prolonged. Left lumbar-to-scalp interpeak latency borderline prolonged.	Performed at 5 months: UE SSEP: Not performed. LE SSEP: No reliable information obtained.	Performed at 17 months: UE SSEP: Not performed. LE SSEP: Normal bilateral absolute lumbar potentials. No other reliable information obtained.	No reliable interpretation.
Patient 6	UE SSEP: Right clavicle-to-cervical and left clavicle-to-scalp interpeak latency mildly prolonged. LE SSEP: Normal bilateral lumbar potentials. Normal left cervical absolute and lumbar-to-cervical interpeak latency. No other reliable information obtained.	Performed at 5 months: UE SSEP: Left clavicle-to-scalp interpeak latency borderline prolonged. Normal right absolute and interpeak latencies. LE SSEP: Normal left cervical and bilateral lumbar absolute latencies. Normal lumbar-to-cervical interpeak latency. No other reliable information obtained.	Performed at 17 months: UE SSEP: Left clavicle-to-scalp interpeak latency borderline prolonged. Normal right absolute and interpeak latencies. LE SSEP: Normal bilateral lumbar potentials. No other reliable information obtained.	No significant change from baseline.
Patient 7	UE SSEP: Normal. LE SSEP: No reliable information obtained.	Performed at 7 months: UE SSEP: Not performed. LE SSEP: Normal bilateral lumbar absolute latencies. No other reliable information obtained.	Performed at 17 months: UE SSEP: Not performed LE SSEP: Normal bilateral absolute lumbar latencies. Right absolute scalp and lumbar-to-scalp interpeak latencies prolonged.	No reliable interpretation.
Patient 8	UE SSEP: Left cervical and scalp absolute latencies prolonged. All left interpeak latencies prolonged. Right clavicle-to-scalp interpeak latency prolonged. LE SSEP: No reliable information obtained.	Performed at 11 months: UE SSEP: Bilateral clavicle-to-cervical and clavicle-to-scalp interpeak latencies prolonged. LE SSEP: No reliable information obtained.	Performed at 16 months: UE SSEP: Bilateral clavicle-to-scalp interpeak latencies prolonged. Right clavicle-to-cervical interpeak latency prolonged. LE SSEP: Normal left cervical and right lumbar absolute latencies. No other reliable information obtained.	No change from baseline.

Patient 9	UE SSEP: Bilateral clavicle-to-scalp and right clavicle-to-cervical interpeak latencies prolonged. LE SSEP: Bilateral lumbar-to-scalp interpeak latency prolonged.	Performed at 7 months: UE SSEP: Normal LE SSEP: Left absolute scalp latency normal. No other reliable information obtained.	Performed at 15 months: UE SSEP: Normal LE SSEP: Normal bilateral absolute scalp latencies. No other reliable information obtained.	Resolution of upper extremity central potential prolongation. No reliable interpretation for lower extremities.
Patient 10	UE SSEP: Normal right absolute clavicle latency. No other reliable information obtained LE SSEP: No reliable information obtained	NA	NA	No reliable interpretation.
Note: All upper extremity studies were performed via stimulation of the median nerve at the wrist, and all lower extremity studies were performed via stimulation of the tibial nerve at ankle.				
Abbreviations: UE SSEP , upper extremity somatosensory evoked potential with median nerve (wrist) stimulation; LE SSEP , lower extremity somatosensory evoked potential with tibial nerve (ankle) stimulation; NA , not available				

6) Did changes in the sensory levels occur?

Response to reviewer: We kindly thank the reviewer for their comment. Nine patients displayed some degree of improvement in sensation in at least one level. Similarly, seven patients experienced improvement in motor function in at least one level. This information is reflected in Tables 5 and 6. In addition, the text has been modified accordingly to provide more information on these changes.

Change to text: Lines 231-236: "In addition, seven patients displayed improvement in motor and sensory function in at least one level, while two patients experienced improvement in sensation in at least one level. **Table 5** represents the ISNCSCI score changes for Motor, Sensory – Pin Prick, and Sensory – Light Touch from baseline at weeks 4, 24, 48, and 96. **Table 6** shows the number of ISNCSCI muscle function levels and dermatomes improved per patient at the final follow-up for Motor, Sensory – Pin Prick, and Sensory – Light Touch."

Reviewer #2 (Remarks to the Author):

This is a well-conducted Phase I trial on the Biological and Clinical Responses to Intrathecal Delivery of Autologous Adipose-Derived Mesenchymal Stem Cells in Patients with Traumatic Spinal Cord Injury.

The study design, methods, and presentation (tables and figures) and interpretation of the results are mostly adequate.

However, there is one issue in trial design needing attention. A typical phase I design is 3+3 with dose escalation, but we didn't see this in this phase I trial. Why was the dosage of 100 million cells chosen? Why the trial is a single dosage trial and didn't escalate dosage as there is no SAE (toxicity)?

Response to reviewer: We thank their reviewer for their comment. The current study's dosage regimen was guided by a preceding Phase I dose-escalation trial that assessed the safety of intrathecal injections of AD-MSCs in individuals with amyotrophic lateral sclerosis. The reference trial conducted by Staff et al. 2016 utilized the same cellular product as the one being evaluated in our study. Their outcomes indicate a favorable safety profile at the highest administered dose of 100 million cells in a single administration. We have modified the methods accordingly to provide the reasoning behind the trial's design and dose selection.

Change to text: Lines 106-113: "The dosage protocol employed in the present study was informed by a precedent Phase I dose-escalation trial, which studied the safety of intrathecal AD-MSCs in amyotrophic lateral sclerosis (ALS).¹⁷ The reference trial utilized the same cellular product as the one under evaluation in our current study. It revealed a positive safety profile at the highest dose of a single dose of 100 million cells. The present study was carried out as a single-dose Phase I trial where all ten patients were closely followed for the occurrence of any adverse events (AEs)."

Reviewer #3 (Remarks to the Author):

The manuscript by Bydon et al. entitled "Biological and clinical responses to intrathecal delivery of autologous adipose-derived mesenchymal stem cells in patients with traumatic spinal cord injury: a phase 1 trial" describes the clinical outcomes in 10 spinal cord injury patients enrolled in a phase 1 clinical trial evaluating the intrathecal administration of autologous adipose tissue-derived mesenchymal stem cells. Of the ten enrolled patients, six had a cervical-level injury, and four had a thoracic-level injury. All were enrolled within 12 months of injury and had injuries classified as AIS A or AIS B.

The data presented is a look at a larger dataset from the CELLTOP trial, from which data on a single patient had been published previously. Overall, the manuscript presents noteworthy results found in the patients in the trial. Those described were that autologous AD-MSCs intrathecal infusions were well

tolerated with minimal adverse events. More notably, 7 of the ten patients had improvements in their AIS grade from the time of injection, with three demonstrating noticeable improvements in motor and sensory function. The report on this small number of patients could be important in the SCI treatment field, though similar trials in SCI have presented mixed results.

While the data looks very promising, the reviewer has several significant issues that reduce the enthusiasm for the manuscript as presented.

It is unclear whether the data on the previously published patient is part of the current dataset or are these ten distinct patients.

Response to reviewer: We thank the reviewer for their comment. The patient presented in the previously published case report is included in the current cohort of 10 patients. We have adjusted the text in the introduction to provide further clarity.

Change to text: Lines 91-92: "Data from one of the participants included in this study have been previously published as a case report.¹⁶"

The Materials and Methods as presented are insufficiently detailed, so the work could not be reproduced as prepared.

The section on the isolation and preparation of AD-MSCs is woefully inadequate. They refer to a previous publication (Singer et al., 2019) to address cell preparation. However, the previous publication has no information on cell isolation, characterization and testing. As prepared, the reader is not indicated what they were administering to patients.

Response to reviewer: We thank the reviewer for their insight. The stem cell isolation and preparation implemented in our study was based on that employed in the Singer et al. 2019 study. Instead of only referring to the study, we have included a detailed explanation of the preparation, isolation, expansion, and release testing processes utilized in our trial, based on the DOSE guidelines.

Change to text: Lines 123-133 "The following description of the cell therapy product is based on the DOSES guidelines.¹⁹ The cell-manufactured product was an autologous derived from adipose tissue and purified by culture flask adherence. Cells were expanded in 5% platelet lysate-based media (PLTMax; Mill Creek LifeSciences, Rochester, MN). Cells were expanded until sufficient cell dosage was achieved and cryopreserved for release testing. Release testing consisted of sterility tests (mycoplasma, aerobic and

anaerobic culture growth), genetic stability (karyotype analysis), and purity and potency assays using flow cytometry as recommended.²⁰ Isolation, expansion, and release testing was completed 4-6 weeks after AD-MSC isolation from the adipose tissues. Upon release, the cells were thawed and allowed to recover for up to four days in media. On the day of treatment, cells were collected, washed, and resuspended in lactated Ringer's solution for intrathecal delivery at the designated dose of 100 million cells in 10 ml."

Moreover, the cell dosing is not described at all. There is a cursory mention of cell dose in the Discussion. Later in the Discussion, the authors mention this being a dose escalation study; however, it is not described in detail, which is completely unacceptable. Which patients got a single dose versus multiple doses? Were the multiple-dose patients the ones who saw marked improvements? It is never discussed.

Response to reviewer: We kindly thank the reviewer for their comment. The section that the reviewer is referring to in the discussion, refers to the study conducted by Staff et al. 2016, who conducted a Phase I dose-escalation trial investigating the safety of intrathecal administration of the AD-MSC in ALS. Our study employed a single-dose protocol and none of the patients received multiple doses. To ensure clarity, the text has been modified to highlight that the study conducted by Staff et al. implemented a dose-escalation protocol and not ours.

Change to text: Lines 106-113: "The dosage protocol employed in the present study was informed by a precedent Phase I dose-escalation trial, which studied the safety of intrathecal AD-MSC injections in amyotrophic lateral sclerosis (ALS).¹⁷ The reference trial utilized the same cellular product as the one under evaluation in our current study. It revealed a positive safety profile at the highest dose of a single dose of 100 million cells. The present study was carried out as a single-dose Phase I trial where all ten patients were closely followed for the occurrence of any adverse events (AEs)."

Lines 264-267: "The safety profile has also been corroborated in a prior study with amyotrophic lateral sclerosis (ALS) conducted by Staff et al. 2016. In contrast to our study, Staff et al. involved a dose-escalation protocol ranging from 1×10^7 (single dose) to 1×10^8 cells (2 monthly doses)."

Also, several figures mentioned in the manuscript do not align with the actual figures in the paper. There does not appear to be Figure 1B or 2B, as mentioned, which causes the other figures to mislabel.

Response to reviewer: We kindly thank the reviewer for their comment. The text has been modified in several places to reflect the correct numbering of tables and figures.

Change to text: Lines 112-113: "Follow-up visits were scheduled on Day 2, Day 3, Week 1, Week 2, Week 6, Week 12, Week 24, Week 48, and Week 96 (**Figure 1**)."

Lines 146-147: "The process of fat harvest, AD-MSCs isolation, culture, and administration is illustrated in **Figure 2**."

Lines 202-203: "**Table 2** depicts the full safety profile for the ten patients."

Lines 208-209: "**Table 3** highlights the MRI changes for all patients before and after stem cell injection."

Line 211-212: "The SSEP findings and interpretations are highlighted in **Table 4**."

Lines 225-226: "Changes in the AIS grade at the time of injury, enrollment, injection, and follow-up are highlighted in **Figure 4**."

Lines 245-246: "These changes are depicted in **Figure 5**."

I also feel that the authors failed to give a thoughtful summary and discussion of their data.

My overall assessment is that this manuscript has been carelessly assembled, which causes the reviewer to have serious concerns about its science.

Response to reviewer: We thank the reviewer for their comment. Taking into consideration the reviewers' comments, the text has been revised to provide a better presentation of our results and a more comprehensive discussion of our findings. We have provided more information regarding our methodology and outcomes and have reflected upon our findings. These changes have been implemented throughout the text, but the key modifications are outlined below.

Change to text: Lines 211-223: "Two patients did not have any change from baseline, an interpretation was not possible for four patients, and one patient had normal SSEPs throughout the study period. The SSEP findings and interpretations are highlighted in **Table 4**. Overall, the SSEP findings, when there was sufficient data for interpretation, correlated with the patients' clinical assessment. In that regard, patients 3 and 4 are worth of particular note. Patient 4, who was AIS grade A at the time of injury, displayed normal SSEPs throughout the study period. The baseline SSEP was performed just prior to the time of injection, when patient 4 was AIS grade C at the time of assessment, which is in keeping with the normal SSEP findings. Regarding patient 3, according to their baseline clinical examination, light touch was not affected as much as pinprick. In addition, this patient's AIS grade B is driven by the complete loss

of motor function below the level of injury. Taking into account that SSEP mainly evaluates dorsal column function, which mediates light touch sensation (whereas pin prick is transferred predominantly through the spinothalamic tracts),^{22,23} the SSEP findings are in keeping with the ISNCSCI examination and the changes seen at follow-up.”

Lines 230-235: "The basis for the AIS changes for each patient is included in **Table 1**. In addition, seven patients displayed improvement in motor and sensory function in at least one level, while two patients experienced improvement in sensation in at least one level. **Table 5** represents the ISNCSCI score changes for Motor, Sensory – Pin Prick, and Sensory – Light Touch from baseline at weeks 4, 24, 48, and 96. **Table 6** shows the number of ISNCSCI muscle function levels and dermatomes improved per patient at the final follow-up for Motor, Sensory – Pin Prick, and Sensory – Light Touch.”

Lines 271-286: “In the present study, a significant proportion of patients exhibited varying degrees of cauda equina thickening, clumping, or nodular enhancement. The existing body of literature presents conflicting findings regarding the association of these imaging characteristics following intrathecal administration of stem cells with neurological deterioration, with some studies suggesting a link, while others failing to establish such a correlation. Notably, case reports have postulated that underlying inflammatory processes triggered by intrathecal stem cell administration may result in nerve root compression and subsequent symptom development.²⁹⁻³¹ Conversely, Singer et al. in their phase I/II trial employed a dose escalation design to assess the intrathecal administration of AD-MSCs in multiple system atrophy and revealed similar MRI findings in all patients within the medium (2 doses of 5×10^7 cells) and high-dose groups (2 doses of 1×10^8 cells).¹⁸ Their proposition centers on these MRI changes being reflective of a reactive response to the stem cell infusion, with more severe instances resembling features of reactive arachnoiditis. Importantly, their patients did not exhibit significant neurological deficits, albeit half of them experienced low-back discomfort. Likewise, the MRI changes observed in our patient cohort did not appear to correlate with neurological manifestations, thereby supporting a more benign interpretation of these findings. This intriguing observation remains an active area of research, and future studies incorporating pathological correlates are imperative to provide a definitive resolution to this question.”

Lines 296-301: “However, the absence of controls prevents ascribing the observed neurological improvement solely to the administration of AD-MSCs. Furthermore, it's essential to consider that the patients included in the study were treated at a quaternary academic center with a highly specialized multidisciplinary team, which likely provided rehabilitation support that differs from what most spinal cord injury patients typically receive. As a result, there is a need for future, larger double-blinded,

randomized controlled trials to investigate the potential benefits of AD-MSC injections in late-stage neurological recovery for SCI.”

Reviewer #4 (Remarks to the Author):

This report is on the CELLTOP phase I study which included tetra and paraparetic spinal cord injured patients receiving adipose tissue-derived mesenchymal stem cells (AD-MSC) 8-22 months after SCI. This work represents a considerable effort and is one of the very few interventional studies in the field of spinal cord injury, using state of the art methodology techniques (incl SSEP). The results demonstrate safety and expectable transient AEs. The reported improvement may just represent spontaneous recovery (plus placebo-effect), being unrelated to intervention itself. Overall, I think this is relevant report, but after incorporation of the suggestions for revision, it should be more suitable for specialized journals (some examples are listed below).

Phase 1 Safety Trial of Autologous Human Schwann Cell Transplantation in Chronic Spinal Cord Injury. Gant KL, Guest JD, Palermo AE, Vedantam A, Jimsheleishvili G, Bunge MB, Brooks AE, Anderson KD, Thomas CK, Santamaria AJ, Perez MA, Curiel R, Nash MS, Saraf-Lavi E, Pearse DD, Widerström-Noga E, Khan A, Dietrich WD, Levi AD. *J Neurotrauma*. 2022 Feb;39(3-4):285-299. doi: 10.1089/neu.2020.7590. Epub 2021 May 3.

Safety of Autologous Human Schwann Cell Transplantation in Subacute Thoracic Spinal Cord Injury. Anderson KD, Guest JD, Dietrich WD, Bartlett Bunge M, Curiel R, Dididze M, Green BA, Khan A, Pearse DD, Saraf-Lavi E, Widerström-Noga E, Wood P, Levi AD. *J Neurotrauma*. 2017 Nov 1;34(21):2950-2963. doi: 10.1089/neu.2016.4895. Epub 2017 Mar 21.

Emerging Safety of Intramedullary Transplantation of Human Neural Stem Cells in Chronic Cervical and Thoracic Spinal Cord Injury. Levi AD, Okonkwo DO, Park P, Jenkins AL 3rd, Kurpad SN, Parr AM, Ganju A, Aarabi B, Kim D, Casha S, Fehlings MG, Harrop JS, Anderson KD, Gage A, Hsieh J, Huhn S, Curt A, Guzman R. *Neurosurgery*. 2018 Apr 1;82(4):562-575. doi: 10.1093/neuros/nyx250.

Minor

1. Abstract: More important than the timepoint when patient has been enrolled is when the patient has received the LP (receiving intervention).

Response to reviewer: We thank the reviewer for their suggestion. The text in the abstract has been modified accordingly to include the mean time and range for the time from injury to injection.

Change to text: Lines 49-50: "All patients were enrolled within 12 months of injury (range: 2-12 months), with a mean time from injury to injection of 12 months (range 7 – 22 months)."

2. Figure 3: Y-axis label in Figure 3 of the axis should be "AIS Grade" (not 'ASIA Scores').

Response to reviewer: We thank the reviewer for their comment. The Y-axis label has been corrected accordingly to "ASIA Impairment Scale (AIS) Grade" and the legend title has been changed to "Grades of AIS increase."

Change to Figure 4: Line 482-484

3. AD-MS-C were propagated to 100 Million cells. Was this also the amount of cells injected into the patient? How large was the volume injected into the CSF?

Response to reviewer: We thank the reviewer for their insightful comment. All patients were injected with 100 million cells as per the study's protocol. The overall volume injected in the CSF at the time of injection was 10 ml.

Change to text: Lines 132-133: "On the day of treatment, cells were collected, washed, and resuspended in lactated Ringer's solution for intrathecal delivery at the designated dose of 100 million cells in 10 mls."

REVIEWER COMMENTS

Reviewer #1 (Remarks to the Author):

The authors have improved the manuscript. However, tables 5 and 6 are difficult to understand. What is average motor change per time point for an individual subject? In Table 5 are the scores total score changes, or changes from baseline? Patient 9, in particular, is confusing- is there a loss of 24 sensory points at endpoint? What is the interpretation of that regarding safety? How does this reconcile with what is shown in Table 6?

Is Table 6 intended to illuminate the change in the NLI?

It would be much better to illustrate these changes with a figure showing dermatomal and myotomal changes on the ISNCSCI body map, as others have done.

Reviewer #2 (Remarks to the Author):

Thanks authors for their effort to improve the manuscript. I am satisfied with the response and revision. No further issues needing attention.

Reviewer #3 (Remarks to the Author):

The authors have effectively addressed the concerns raised in my review. Overall, the manuscript is a more effective presentation of their data, findings and the impact.

REVIEWER COMMENTS

Reviewer #1 (Remarks to the Author):

The authors have improved the manuscript. However, tables 5 and 6 are difficult to understand. What is average motor change per time point for an individual subject? In Table 5 are the scores total score changes, or changes from baseline? Patient 9, in particular, is confusing- is there a loss of 24 sensory points at endpoint? What is the interpretation of that regarding safety? How does this reconcile with what is shown in Table 6?

Response to Reviewer: We thank the reviewer for their comment. Table 5 represents the difference in each International Standards for Neurological Classification of Spinal Cord Injury (ISNCSCI) domain (Motor, Light Touch, Pin – Prick) between each follow-up and baseline. The change is calculated by subtracting the total baseline score for each respective domain from the total score at each follow-up. It is intended to reflect the magnitude of improvement/deterioration for each domain at each follow-up. The term “average” was used to convey that the calculated values consider both upper and lower extremity scores. However, as highlighted by the reviewer's comment, this description may be misleading; thus, Table 5 has been modified to avoid confusion. Table 6 represents the number of levels improved and the degree of improvement from baseline at the final follow-up. The values in Table 6 are calculated by counting the total number of levels that improved at final follow-up and also the total number of levels that improved by 1 or 2 for the sensory domains and by 1-5 for the motor domain. Table 6 has been updated accordingly to improve clarity. We have also included supplementary material with a dermatomal map for each patient, representing the sensory scores on ISNCSCI examination at baseline and final follow-up.

The reviewer also raises an interesting point regarding Patient 9. At the final follow-up, Patient 9 experienced improvement in the motor score (by a total of 26 points), deterioration in pinprick (by a total of 11 points), and deterioration in light touch (by a total of 23 points). It is noteworthy that Table 5, reporting the total change, and Table 6, reporting the number of levels that improved, present divergent values as they reflect different elements of the ISNCSCI examination. This is better demonstrated in the supplementary file, which includes detailed dermatomal body maps. Furthermore, this pattern of change was not noted in other patients. Considering the study design and the absence of controls, a definitive link between the observed changes and stem cell administration is challenging. This point is highlighted in the discussion (Lines 290-291: “However, the absence of controls prevents ascribing the observed neurological improvement solely to the administration of AD-MSCs.”).

Change to text: Lines 229-232: “Table 5 represents the ISNCSCI score changes from baseline for Motor, Sensory – Pin Prick, and Sensory – Light Touch at weeks 4, 24, 48, and 96. Table 6 shows the number of ISNCSCI muscle function levels and dermatomes improved per patient at the final follow-up (compared to baseline) for Motor, Sensory – Pin Prick, and Sensory – Light Touch.”

Lines: 446-449: Table 5: International Standards for Neurological Classification of Spinal Cord Injury (ISNCSCI) Motor, Sensory – Pin Prick, and Sensory – Light Touch score changes from baseline at weeks 4, 24, 48, and 96.

Patient #	Total Motor Change from Baseline				Total Pin Prick Change from Baseline				Total Light Touch Change from Baseline			
	Week 4	Week 24	Week 48	Week 96	Week 4	Week 24	Week 48	Week 96	Week 4	Week 24	Week 48	Week 96
Patient 1	13	18	21	25	18	25	22	26	10	17	25	43
Patient 2	0	1	1	1	0	1	-4	0	1	1	-3	1
Patient 3	1	-5	-6	1	23	14	19	29	19	10	5	11
Patient 4	5	4	2	0	10	7	-9	-3	-1	0	-1	-6
Patient 5	-1	-3	-2	-3	-4	-2	-2	-4	-3	-2	-3	-4
Patient 6	-1	2	-1	1	9	6	7	6	8	5	2	3
Patient 7	0	0	0	2	0	-1	0	1	0	0	3	0
Patient 8	-1	-2	-1	-2	4	2	5	1	2	2	0	-3
Patient 9	27	24	34	26	1	-8	11	-11	-7	-9	-4	-13
Patient 10	4	8	11	12	5	-1	-6	3	4	-3	-6	7

Note - The values are derived by summing the scores in each domain (Motor, Pin Prick, Light Touch) for the upper and lower extremities. Subsequently, the baseline score is subtracted from the corresponding follow-up score for each respective domain.

Is Table 6 intended to illuminate the change in the NLI?

Response to Reviewer: We thank the reviewer for their comment. As described above, Table 6 represents the number of levels that improved at final follow-up compared to baseline. It also includes the total number of levels that improved by 1 or 2 for Light Touch and Pin Prick scores, and by 1-5 for the Motor scores. It is intended to reflect the total number of levels improved and the respective magnitude of changes. A note has been added to Table 6 to improve clarity.

Change to Text: Lines 450-454: Table 6: Number of International Standards for Neurological Classification of Spinal Cord Injury (ISNCSCI) muscle function levels and dermatomes improved at the final follow-up for Motor, Sensory – Pin Prick, and Sensory – Light Touch.

Patient #	Motor Change Improvement					Pin Prick Improvement			Light Touch Improvement			
	Number of spinal levels improved	Spinal levels improved by 1	Spinal levels improved by 2	Spinal levels improved by 3	Spinal levels improved by 4	Spinal levels improved by 5	Number of spinal levels improved	Spinal levels improved by 1	Spinal levels improved by 2	Number of spinal levels improved	Spinal levels improved by 1	Spinal levels improved by 2
Patient 1	19	13	6				24	22	2	40	38	2
Patient 2	1	1								1	1	
Patient 3	1	1					28	27	1	11	11	
Patient 4							3	3		1	1	
Patient 5												
Patient 6	1	1					5	1	4	5	4	1
Patient 7	2	2					1	1				
Patient 8							2	1	1	2	1	1
Patient 9	12	5	2	4		1	2	2		3	3	
Patient 10	8	5	2	1			2	1	1	7	7	

Note - The values represent the number of spinal levels that improved on ISNCSCI examination at the final follow-up compared to baseline.

It would be much better to illustrate these changes with a figure showing dermatomal and myotomal changes on the ISNCSCI body map, as others have done.

Response to Reviewer: We thank the reviewer for their suggestion. We have added dermatomal maps reflecting sensory level scores at baseline and final follow-up for all patients, following the ISNCSCI format.

Change to text: Supplementary Material File 2 - Dermatomal Body Maps

Reviewer #2 (Remarks to the Author):

Thanks authors for their effort to improve the manuscript. I am satisfied with the response and revision. No further issues needing attention.

Response to Reviewer: We thank the reviewer for their time and effort in reviewing our manuscript.

Reviewer #3 (Remarks to the Author):

The authors have effectively addressed the concerns raised in my review. Overall, the manuscript is a more effective presentation of their data, findings and the impact.

Response to Reviewer: We thank the reviewer for their time and effort in evaluating our manuscript.

REVIEWERS' COMMENTS

Reviewer #1 (Remarks to the Author):

The manuscript has been improved. The addition of the sensory maps is beneficial, allowing correlation to other studies. It should be mentioned why the particular cytokine panel was used since it does not capture common inflammatory markers such as IL-6, IL-10, and TNF-alpha.

Did the authors capture SCIM scores or QOL measures?

For SSEPs, should the lumbar not be positive in all subjects as a control for the stimulus inputs?

REVIEWERS' COMMENTS

Reviewer #1 (Remarks to the Author):

The manuscript has been improved. The addition of the sensory maps is beneficial, allowing correlation to other studies. It should be mentioned why the particular cytokine panel was used since it does not capture common inflammatory markers such as IL-6, IL-10, and TNF-alpha.

Response to Reviewer: We thank the reviewer for their comment. The purpose CSF cytokine analysis was to explore the biological effect of AD-MSc treatment. We included VEGF-A, TNF- β , GM-CSF with other pro-inflammatory and angiogenic cytokines with known implications to gain insight into the mechanism of regeneration. Elevated levels of IL-6, IL-10, and TNF - α have been predominantly observed in the acute post-injury phase, with some evidence suggesting lasting effects in the intermediate and late post-injury stages. Although the cytokine panel utilized in our study assessed various pro-inflammatory and angiogenic cytokines that have been linked to spinal cord injury, expanding the assessment to include other known inflammatory markers, such as IL-6, IL-10, and TNF- α , could potentially complement findings. Consequently, it would be worthwhile to investigate these cytokine levels in future trials, including our ongoing Phase 2 trial. The manuscript has been modified to note the utility of expanding the assessment of cytokines in future studies.

Change to Manuscript: Lines 198-202: "Furthermore, a variety of cytokines, such as IL-6, IL-10, and TNF- α , which were not included in the cytokine panel utilized in our study, have been implicated in the early and late post-injury phases. Thus, it would be worthwhile for future studies to expand the assessment of cytokines and encompass a more extensive array of immunomodulatory and angiogenic markers."

Did the authors capture SCIM scores or QOL measures?

Response to Reviewer: We thank the reviewer for their comment. Spinal Cord Independence Measure (SCIM) scores and Quality of Life (QOL) metrics were not collected during the trial. Clinical outcomes were captured as part of the International Standards for Neurological Classification of Spinal Cord Injury (ISNCSCI) examination. As a Phase I trial, the primary focus of the index study was evaluating the safety and biological response of adipose tissue-derived mesenchymal stem cells (AD-MSc) in spinal cord injury (SCI), and as such clinimetric tools assessing efficacy were not the focus of the study. The manuscript has been modified to highlight the need for further investigation regarding the efficacy of AD-MSc in SCI.

Change to Manuscript: Lines 191-193: "As a result, there is a need for future, larger randomized controlled trials employing various clinimetric tools and quality of life measures to investigate the potential benefits of AD-MSc injections in late-stage neurological recovery for SCI."

For SSEPs, should the lumbar not be positive in all subjects as a control for the stimulus inputs?

Response to Reviewer: We thank the reviewer for their note. It is worth noting that Somatosensory Evoked Potentials (SSEPs) are specific to dorsal column function, which mediates light touch and proprioception. Consequently, they reflect a specific part of sensation, and their findings may not correlate with the ISNCSCI examination of patients who exhibit primarily spinothalamic deficits that carry pin-prick sensations but have otherwise normal light touch sensation. This is seen, for example, in

patient 3, as it is highlighted in the manuscript. Consequently, the lumbar potentials would not be expected to be uniform throughout the cohort of patients and are primarily dependent on each patient's underlying injury.

Change to Manuscript: None